# Diversity and Differentiation of Duckweed Species from Israel

**DOI:** 10.3390/plants11233326

**Published:** 2022-12-01

**Authors:** Avital Friedjung Yosef, Lusine Ghazaryan, Linda Klamann, Katherine Sarah Kaufman, Capucine Baubin, Ben Poodiack, Noya Ran, Talia Gabay, Shoshana Didi-Cohen, Manuela Bog, Inna Khozin-Goldberg, Osnat Gillor

**Affiliations:** 1Zuckerberg Institute for Water Research, J. Blaustein Institutes for Desert Research, Ben Gurion University, Midreshet Ben-Gurion 8499000, Israel; 2Department of Ecology and Evolutionary Biology, University of Colorado Boulder, Boulder, CO 80309, USA; 3Department of Life Sciences, Ben-Gurion University of the Negev, Be’er Sheva 8410501, Israel; 4French Associates Institute for Agriculture and Biotechnology of Drylands, The Jacob Blaustein Institutes for Desert Research, Ben-Gurion University of the Negev, Midreshet Ben-Gurion 8499000, Israel; 5Institute of Botany and Landscape Ecology, University of Greifswald, 17489 Greifswald, Germany

**Keywords:** duckweed, fatty acids, DNA barcoding, diversity, biogeography, nitrogen content, protein concentration, migration

## Abstract

Duckweeds (Lemnaceae) are tiny plants that float on aquatic surfaces and are typically isolated from temperate and equatorial regions. Yet, duckweed diversity in Mediterranean and arid regions has been seldom explored. To address this gap in knowledge, we surveyed duckweed diversity in Israel, an ecological junction between Mediterranean and arid climates. We searched for duckweeds in the north and center of Israel on the surface of streams, ponds and waterholes. We collected and isolated 27 duckweeds and characterized their morphology, molecular barcodes (*atpF*-*atpH* and *psbK*-*psbI*) and biochemical features (protein content and fatty acids composition). Six species were identified—*Lemna minor*, *L. gibba* and *Wolffia arrhiza* dominated the duckweed populations, and together with past sightings, are suggested to be native to Israel. The fatty acid profiles and protein content further suggest that diverged functions have attributed to different haplotypes among the identified species. *Spirodela polyrhiza*, *W. globosa* and *L. minuta* were also identified but were rarer. *S. polyrhiza* was previously reported in our region, thus, its current low abundance should be revisited. However, *L. minuta* and *W. globosa* are native to America and Far East Asia, respectively, and are invasive in Europe. We hypothesize that they may be invasive species to our region as well, carried by migratory birds that disperse them through their migration routes. This study indicates that the duckweed population in Israel’s aquatic environments consists of both native and transient species.

## 1. Introduction

The Lemnaceae (duckweeds) family comprises the world’s smallest and fastest growing seed plants [1]. The duckweeds are miniscule plants that float on or below the surface of freshwater bodies. The duckweed family was found throughout the globe, except for polar regions, and was classified into five genera with 36 species [2]. Representatives of three genera contain one or few tiny roots emerging from the fronds (*Spirodela, Landoltia* and *Lemna*), while the two remaining genera are rootless and smaller (*Wolffiella* and *Wolffia*) [1]. These diminutive plants have gone through an extreme reduction in body size, with some species less than 0.5 mm in size, thereby minimizing the need for non-photosynthetic organs and selecting for rapid multiplication through budding [3]. As a consequence of their fast growth rate, biomass production is high, providing practical applications to the duckweeds in food [4,5], feed [6], water treatment [7,8] and biotechnology [9,10].

The first attempts to classify duckweed were based on their morphology [1]. However, due to the duckweed’s diminutive size and organ reduction, morphological and anatomical classifications are challenging. Therefore, over the years, there was an attempt to classify duckweeds based on the chemical composition of the flavonoids [11], isoforms of enzymes (allozymes) [12] and fatty acids [13]. With the advancement of molecular taxonomy, molecular methods of identification have been developed, including molecular fingerprinting and sequencing [14]. The DNA-based molecular identification is based on the polymorphisms of target non-coding intron and gene spacer regions, mainly within the chloroplast genome [15,16]. DNA markers are considered the most reliable method for species classification and have been demonstrated to be capable of detecting polymorphisms among haplotypes of the same species [2]. The molecularly detected polymorphisms are supported by different physiological properties, including growth rate [3], protein and starch content [17], metabolite abundance [13] and turion formation [18]. Therefore, haplotype identification requires a combination of molecular and physiological-based methods [19].

Reliable identification methods are vital to establish the biogeography of duckweeds. Early studies have reported species-dependent biodiversity, ranging from regional to global distribution, with some species showing a broad distribution, while others were restricted to certain regions [16]. Yet, a species’ global dispersal could result from the generalization of unique haplotypes within a species. In fact, it was reported that dispersal was not directly linked to taxonomy as closely related species—even haplotypes of the same species—were detected oceans apart [16,20].

Duckweed growth is mostly vegetative, whereas flowering and seed generation, i.e., sexual reproduction, is rarer [21]. The survival of duckweeds during winter in temperate regions is not only dependent on seeds; this is further evidenced by various duckweeds reportedly sinking to the bottom of water bodies, consequently morphing into turions that can survive freezing [18]. Moreover, dispersal was also attributed to biotic vectors, for example via migrating waterfowl that can carry the small duckweeds over great distances in their gastrointestinal tract or attached to their body [22,23]. When introduced to new aquatic habitats, the duckweed’s fast vegetative growth facilitates their propagation and establishment [23].

The study of duckweed diversity in the context of biogeographical distribution is relevant to Israel because it is a meeting point between three continents: Africa, Asia and Europe, thereby forming a transitional region between arid and Mediterranean climates [24]. Moreover, the Jordan valley in the east of Israel is extended from the African Rift and serves as an important hub for migratory birds that winter in Africa and summer in Europe [25]. It is a part of the Afro-Palaearctic bird migration system, the largest land bird migration system in the world [26]. In spite of its small size, Israel’s location between the Mediterranean Sea in the west and the Arabian Deserts in the east forms an ecological corridor and a bottleneck in the birds’ flight path, making it an essential stop-over site during migration. The bird’s stopover sites provide an opportunity for hitchhiking plants, such as duckweeds, to establish in new environments [27]. Nevertheless, the diversity of Israel’s duckweeds was never systematically investigated, though sightings of duckweed have been documented. The following Lemnaceae species have been reported in Israel: in the genus *Lemna L. trisulca*, *L. gibba*, *L. aequinoctialis* and *L. minor*; in the genus *Spirodela*: *S. polyrhiza*; and in the genus *Wolffia*: *W. arrhiza* and *W. globosa* were reported (https://flora.org.il/plants/systematics/lemnaceae/ (accessed on 5 September 2022)). The following species are listed as endangered: *W. arrhiza* and *S. polyrhiza* (https://redlist.parks.org.il/plants/list/ (accessed on 5 September 2022)). The species *W. globosa* was reported but considered an invador. However, the difficulty of identifying duckweeds based solely on their morphology, questions the reliability of these observations.

In this study we conducted a systematic survey of duckweeds in northern and central Israel by following past sightings of duckweeds. This involved sampling the aquatic plants, then isolating them in the lab and identifying them through morphology, molecular methods, as well as biochemical features including fatty acid composition and nitrogen content. We hypothesized that the duckweed diversity in these sites would reflect the species reported in Africa, Europe and Central Asia, following birds’ migration routes.

## 2. Materials and Methods

### 2.1. Survey of Duckweed Strains

During June 2021, duckweed species were collected from ponds, springs, streams and waterholes in northern and central Israel (Galilee, Hula Valley, Golan Heights and Sharon). The survey locations were selected based on previous observations from the last century taken from the Israel Nature and Parks Authority database (https://redlist.parks.org.il/plants/list/ (accessed on 11 September 2022)).

Duckweed plants were detected in 24 of the 67 reported locations detailed in the database. The verified duckweed locations are depicted in Figure 1 and detailed in Appendix A.

### 2.2. Duckweed Collection

Plant samples were collected in duplicates in 100 mL plastic containers. The containers were stored in a refrigerated cooler (~10 °C) and transported to the lab up to 48 h after collection. For each confirmed collection site, the pH value of the water was measured by litmus paper and the results are detailed in Appendix A. In the laboratory, electric conductivity (indicating water salinity) was measured using a conductivity meter (Cole-Parmer EW-19820-10, Vernon Hills, IL, USA) and the results are listed in Appendix A.

### 2.3. Cultivation

In the laboratory, the collected duckweeds were sorted according to their morphology and each isolate was sterilized by rinsing the separated fronds with a 2% sodium hypochlorite solution (NaClO) for 2 min. Single fronds were picked and cultured in 0.5 × Schenk and Hildebrandt (SH) basal salt mixture (Sigma–Aldrich, St. Louis, MI, USA) supplemented by 1% sucrose at pH 5.8. The plants were grown in a controlled climate chamber under 25 °C, 16 h light/8 h dark cycles, and 200–250 µmol m^−2^ s^−1^ light intensity. Sterilization was repeated until a single unique isolate was detected. From each unique isolate, a single sterile frond was retrieved and cultured in 0.5 × SH agar with 0.5% sucrose and supplemented with 100 mg L^−1^ cefotaxime (Sigma) to avoid fungal contamination.

### 2.4. Morphological Identification

The morphology of the isolates was assessed according to Landolt 1986 [1] as well as Les et al. [11]. An M205 FCA fluorescence stereo microscope (Leica, Wetzlar, Germany) and Axio Imager 2 light microscope (Zeiss, Jena, Germany) was used.

### 2.5. DNA Extraction, Fragment Amplification and Sequencing

The isolates were cultivated as described above for 10–14 days then total DNA was extracted using DNeasy Plant Pro kit (Qiagen, Hilden, Germany) following the manufacturer’s instructions. The extracted DNA was used as a template to amplify two plastid barcode loci of noncoding intergenic spacers: *atpF*-*atpH* (5′-ACTCGCACACACTCCCTTTCC-3′ and 5′-GCTTTTATGGAAGCTTTAACAAT-3′) and *psbK*-*psbI* (5′-TTAGCATTTGTT TGGCAAG-3′ and 5′-AAGTTTGAGAGTAAGCAT-3′). The amplification was performed as follows: 95 °C pre-denatured for 3 min, followed by 35 cycles of 95 °C for 45 s, 55 °C for 45 s, 72 °C for 45 s, and a further extension at 72 °C for 10 s. Purification of the resulting amplicons was carried out using the AccuPrep^®^ PCR/Gel Purification Kit (Bioneer, Daejeon, S. Korea), according to the manufacturer’s instructions. The purified PCR fragments were sequenced at McLab (San Francisco, CA, USA).

### 2.6. DNA Barcoding Analysis

DNA sequence alignment was generated using Geneious Prime version 2022.1.1 (http://www.geneious.com/prime/ (accessed on 15 September 2022)). Blast analysis was performed using NCBI database (https://www.ncbi.nlm.nih.gov/ (accessed on 17 September 2022)) and Rutgers database (http://epigenome.rutgers.edu/cgi-bin/duckweed/blast.cgi (accessed on 17 September 2022)). Duckweed species reference sequences of the two loci *atpF*-*atpH* and *psbK*-*psbI* were taken from the NCBI database and added to the tree analysis. Multiple alignments of both loci were performed using MUSCLE Alignment (https://www.ebi.ac.uk/Tools/msa/muscle/ (accessed on 20 September 2022)). A phylogenetic tree was constructed using Geneious Tree Builder using the Neighbor-Joining method with Tamura-Nei as the genetic distance model. Support values were calculated using bootstrapping with 1000 reiterations.

### 2.7. Fatty Acids Analysis

Plant fatty acid composition and content were analyzed using a direct transmethylation procedure. Plants were grown as described above for 14–21 days. After harvesting, the cultures were placed in 20 °C for 12 h, then dried for 48 h in a lyophilizer (VirTis, Gardiner, NY, USA). The dry material was ground (ULTRA-TURRAX, IKA, Merck) for 2 min at 6000 rpm. A total of ~10 mg of freeze-dried biomass was used in duplicate for analysis. Cellular fatty acids were converted into methyl esters (FAMEs) by incubation in 2 mL of 2% H_2_SO_4_ in dry methanol (*v*/*v*) for 1.5 h at 90 °C with continuous stirring under Argon gas atmosphere. Myristic acid (C_17:0_) was used as an internal standard for FAME quantification. The reaction was terminated by the addition of 1 mL of water. A total of 1 mL of Hexane (Sigma) was then added for phase separation and extraction of FAMEs. Hexane fractions were evaporated under N_2_ gas flow and resuspended in 400 µL of hexane. FAMEs were analyzed by gas chromatography coupled with flame ionization detection (GC-FID) on a TRACE Ultra Gas Chromatograph (Thermo Electron, Milan, Italy) equipped with a programmed temperature vaporizing injector, a flame-ionization detector (FID) and a SUPELCO WAX 10 capillary column (L × I.D. 30 m × 0.25 mm, df 0.25 μm, Sigma), using a temperature gradient as follows: 1 min at 130 °C, 8 min of linear gradient to 220 °C and 10 min at 220 °C. Helium was used as a carrier gas. Retention times of FAMEs were compared with those of available commercial FAMEs standards (in-house library), and the literature data [28].

### 2.8. Nitrogen Content Analysis

For the nitrogen content analysis, plants were grown and dried as described above. For each sample, 2–3 mg of dry material was taken in triplicate. A Flash Smart elemental analyzer (OEA 2000, Thermo Fisher Scientific, Waltham, MA, USA) was used for the analysis. The nitrogen concentration results were adjusted to 1 g of dry weight. The estimation of protein was carried out as is commonly achieved by multiplying total nitrogen by the numeric factor 6.25 but with the addition of a correction factor specific to plants [29]. To validate our results, we measured the nitrogen concentrations in the same samples with FT-MIR (Fourier Transform Midinfrared Spectroscopy) described in the Appendix A.

## 3. Results

### 3.1. Distribution of Duckweed Species in Israel

In total, 24 locations in the north and center of Israel were found to have duckweed (Figure 1 and Appendix A). These included 14 locations in the Golan Heights, five in the Hula Valley, two in the Galilee, and three in the Sharon. Plant samples were identified morphologically under a microscope (Table 1). Three species of *Lemna* have been identified: *L. gibba* at highest occurrence (seven observations), *L. minor* (five observations) and *L. minuta* (three observations). Two species of *Wolffia* have been identified: *W. arrhiza* at high occurrence (seven observations) and *W. globosa* at low occurrence (one observation). In addition, the species *S. polyrhiza* was observed once.

### 3.2. DNA Barcoding Analysis

The sequences derived from the two markers *psbK*-*psbI* and *atpF*-*atpH* were used for identifying the species listed in Table 2. Once the sequences had been cleaned, corrected and aligned, they were blast-matched with sequences in the NCBI and Rutgers University databases. The results for species identification, along with their accession numbers, are presented in Table 2. All of the haplotypes were clearly identified as being of the same species, although there was not always a match between the barcodes *atpF*-*atpH* and *psbK*-*psbI*, as seen in the *S. polyrhiza* 19, *W. arrhiza* 56 and all *L. gibba* haplotypes.

#### Phylogenetic Tree

A phylogenetic tree based on *psbK*-*psbI* and *atpF*-*atpH* sequences was constructed after multiple alignment of the sequences (Figure 2) using a total of the 27 haplotypes collected in this study, as well as *W. globosa* “Mankai” (a domesticated duckweed haplotype that was isolated from the Golan Hights, Israel [5]) and additional reference sequences (*S. polyrhiza* 7498, *W. arrhiza* DW35, *W. globosa* 8789, *L. minuta* 5573, *L. gibba* 5504, *L. minor* 7123) taken from the NCBI database (one for each species).

### 3.3. Fatty Acid Analysis

Total fatty acid content measured by GC-FID varied between 2.83 and 6.34% of dry weight across the isolates. There were three major fatty acids that accounted for 80–90% of the total fatty acids: palmitic acid (16:0), linoleic acid (LA, 18:2*n*6) and α-linolenic acid (ALA, 18:3*n*3). The other fatty acids present at lower proportions are listed in Table 3 and Appendix A.

ALA (18:3*n*3) represented approximately 50% of the total fatty acids in most *Lemna* isolates, whereas *Wolffia* species had lower concentrations of this major *n*3 PUFA with a concurrent increase in 18:2*n*6. Accordingly, the ratio 18:3*n*3/18:2*n*6 attained higher values in *Lemna* species. *Spirodela* differed from other duckweed species by the lowest percentage of 18:2*n*6, resulting in the maximal 18:3*n*3/18:2*n*6 ratio. A variety of differences in fatty acid composition between duckweed genera, including *Lemna*, *Spirodela* and *Wolffia*, have already been reported [4,29]. Another apparent difference in this study was the presence of stearidonic acid (SDA,18:4n3), which is the product of a delta-6 (Δ6) desaturation on 18:3*n*3. In this study, all *L. minor* and *L. gibba* isolates featured the presence of SDA at ~2% to above 4% of total FA, as well as the detectable levels of another Δ6 C18 PUFA, γ-linolenic acid (GLA, 18:3*n*6). These data are in line with the presence of the Δ6 desaturase gene in *Lemna* [29] and *Wolffia* [30] species, enabling the biosynthesis of Δ6 C18 PUFA.

### 3.4. Nitrogen Analysis

Nitrogen content varied widely among strains, ranging from 2.2 to 5.4% of dry biomass (Figure 3). There was no clear pattern with respect to the different duckweed species: Nitrogen content appeared to vary depending on the species and even the strain. *W. globosa* “Mankai” had the highest nitrogen content of 5.82%; it translates to a high protein concentration of 25.52%–36.25%. The isolated *W. globosa* 58 showed a lower nitrogen concentration. *W. arrhiza* produced high N concentrations in all five samples: 4.12–5.42% nitrogen, translating to 18.28%–33.87% protein. The species *L. minuta* also produced high results in two samples: between 4.01–4.54% nitrogen, translating to 17.64–28.38% protein. Similar to *W. globosa*, nitrogen values were obtained in a wide range of values in the species *L. minor* and *L. gibba* as well. *S. polyrhiza* yielded an average nitrogen content of 2.95%, which translates into 12.98–18.44% protein (Figure 3). Because of the high variability, the analysis was validated using FT-MIR spectroscopy that yielded similar results (Appendix A).

## 4. Discussion

In a campaign to explore the diversity of duckweed in Israel, we have followed almost 100 years of reports since 1926 (https://redlist.parks.org.il/plants/list/ (accessed on 22 September 2022)). We found duckweeds in ~40% of the seasonal and perennial water bodies explored (confirming 24 of 67 sights). In the confirmed sites, we collected the species, then we transferred them to the lab, there isolating and identifying the duckweeds. The identification was based on combined approaches that included morphology, molecular analyses and the biochemical features of the isolates (Table 1 and Table 3). As a result, six species were confirmed (Figure 1 and Figure 2), with five of the identified duckweed species previously reported in Israel. One of the identified species, *L. minuta*, was collected in Israel for the first time. In this approach, the two barcode regions previously proposed [31], *atpF-atpH* and *psbK-psbI*, have proven to well complement the morphological identification. Interestingly, this geographically limited study shows a pattern that was reported in geographically broader studies. Here, the two identified *Wolffia* species show stronger intraspecific differences between haplotypes than the other species [31], as was shown for the *W. globosa* haplotypes [14] identified by the two chloroplast markers in *Wolffia*. However, we note that the chloroplast markers do not allow for the identification of hybrids, which may well occur between the two species of *L. minor* and *L. gibba*, as recently described [32].

Three of the isolated species dominated the community: *L. gibba*, *L. minor* and *W. arrhiza* (accounting for 31, 17 and 31% of the isolates, respectively) and were previously sighted in Israel (https://biogis.huji.ac.il/heb/home.html (accessed on 25 September 2022)). We suggest that these species form the stable and established population of duckweeds in Israel. Their continuous presence in perennial and seasonal aquatic sites across Israel over the last century supports our hypothesis. Moreover, populations of *L. gibba*, *L. minor* and *W. arrhiza* were previously reported in North Africa, the Middle East and Europe (https://europlusmed.org/ (accessed on 25 September 2022)).

Three of the remaining species identified in this study, *L. minuta*, *W. globosa* and *S. polyrhiza*, were detected at a lower abundance (Figure 1 and Table 1). *S. polyrhiza* is native to the Middle East and Europe [33] and was previously sighted in Israel (https://biogis.huji.ac.il/heb/home.html (accessed on 26 September 2022)). Yet, *S. polyrhiza*’s low abundance detected in our survey (Figure 1 and Table 1) may reflect the species’ growth inhibition by anthropogenic contamination, as was previously described [34,35]. The other two species identified in Israel, *L. minuta* and *W. globosa*, are not native to our region and are considered invasive species in Europe (https://europlusmed.org/ (accessed on 28 September 2022), [36]). *L. minuta* is native to America and is also an invasive species to Europe where it was first sighted in the 1990′s, probably through an accidental introduction [36]. *L. minuta*’s fast vegetative reproduction, high fitness and aggressive competition damages European lake ecosystems by inhibiting local duckweed populations such as *L. minor* [36]. *L. minuta* was spotted in the Hula Valley, a popular overnight break to many migrating waterfowl [25], hence, we hypothesize that the species was inadvertently carried by the migrating birds on their way from Europe to Africa and established there.

The species *W. globosa* is native to the Far East (Thailand, Cambodia and Laos [1]), China [37], and the Indian subcontinent [3]). Like *L. minor*, it was inadvertently introduced to Europe. Although *W. globosa* is the fastest propagating Angiosperm, with a doubling time as low as 72 h [3], it does not aggressively compete with native species but instead resides alongside them in water bodies across Europe [38]. In Israel, *W. globosa* was occasionally sited [5] but its presence in water bodies was transient (we were unable to find the species in the reported sites). The transient presence of *W. globosa* in Israel may suggest that it cannot survive the long Mediterranean summer drought and is reintroduced to various water bodies by birds migrating from Europe.

The molecular phylogeny of the duckweed species converged some isolates, suggesting a common haplotype (Figure 2), yet the biochemical features of species, including fatty acid profiles and nitrogen content, suggested diversification among the haplotypes (Table 3 and Figure 3). The nitrogen concentration results were validated by two independent methods (Elemental Analyzer (Figure 3) and FT-MIR (Appendix A)). The nitrogen contents of the haplotypes were diverse, but not always taxonomically associated. Diverging nitrogen concentrations among duckweeds were likewise reported for various haplotypes of the genera *Wolffia* [4,30] and *Lemna* [39]. There is a possibility that even if laboratory growth conditions are controlled, the optimal growth conditions for each haplotype and species can differ, which may have an effect on the biochemical composition, and specifically, on the nitrogen and protein content [4]. Fatty acid composition was consistent with previous duckweed surveys [15], showing the predominance of ALA (18:3*n*3) and the presence of a considerable amount of SDA (18:4*n*3) mostly in *Lemna* species. Some *Wolffia* species, yet not the ones isolated here, were shown to contain ALA and SDA (*W. australiana* and *W. microscopica* [30]). The presence of SDA, whose biosynthesis requires Δ6-desaturation, is restricted to only a few terrestrial plant families. However, this n-3 PUFA (SDA) widely occurs in cyanobacteria and algae, as well in some duckweeds [4,13,29], indicating the evolutionary radiation of the Δ6-desaturation, and this fatty acid’s possible importance to duckweeds. Concomitant to the diverse nitrogen contents, some haplotypes of the same species showed variability in their fatty acid content, for instance SDA content in *L. gibba* (Table 3).

Our study suggests that both stable and transient duckweed communities are present side-by-side in Israel’s water bodies. Almost 100 years of duckweed sightings suggest that *L. gibba*, *L. minor* and *W. arrhiza* inhabit the perennial water bodies but also survive the summer drought of seasonal ponds. Resilience of duckweed under drought gained little attention in contrast to survival during water freezing [40,41] and should be further explored, especially considering the currently changing climate. Three additional species were identified: *S. polyrhiza*, *L. minuta* and *W. globosa.* While the first was sighted in this region, the two latter are native to America and the Far East, respectively, and considered invasive species in Europe. We hypothesize that these species are transient in Israel, carried by migrating waterfowl on their way from Europe to Africa and established in water bodies as was previously proposed [16]. However, considering the aggressive nature of some invading species (like *L. minuta* [36]), their potential to endanger the fragile community of the endogenous duckweeds in Israel should be considered. However, to validate the community composition of duckweeds in Israel, additional surveys should be conducted covering wider spatial and temporal scales. These surveys should be accompanied by careful isolation of the species followed by a combination of methods to identify both their taxonomy and traits.

## 5. Conclusions

Here, we described the first survey of duckweed ever performed in Israel—a known junction between Mediterranean and arid environments. We isolated 27 duckweed haplotypes and used morphological and molecular approaches to identify them, resulting in six confirmed species. However, independent of the taxonomy, the haplotypes differed in their fatty acid profiles and protein contents. Three of the species were abundant among sites and confirmed by past sightings, thus, were proposed as being native to Israel. The other three species were rarer with two suspected invaders to our region. Thus, future surveys should be conducted to establish the identity and traits of the native duckweed communities in Mediterranean and arid regions.

## Figures and Tables

**Figure 1 plants-11-03326-f001:**
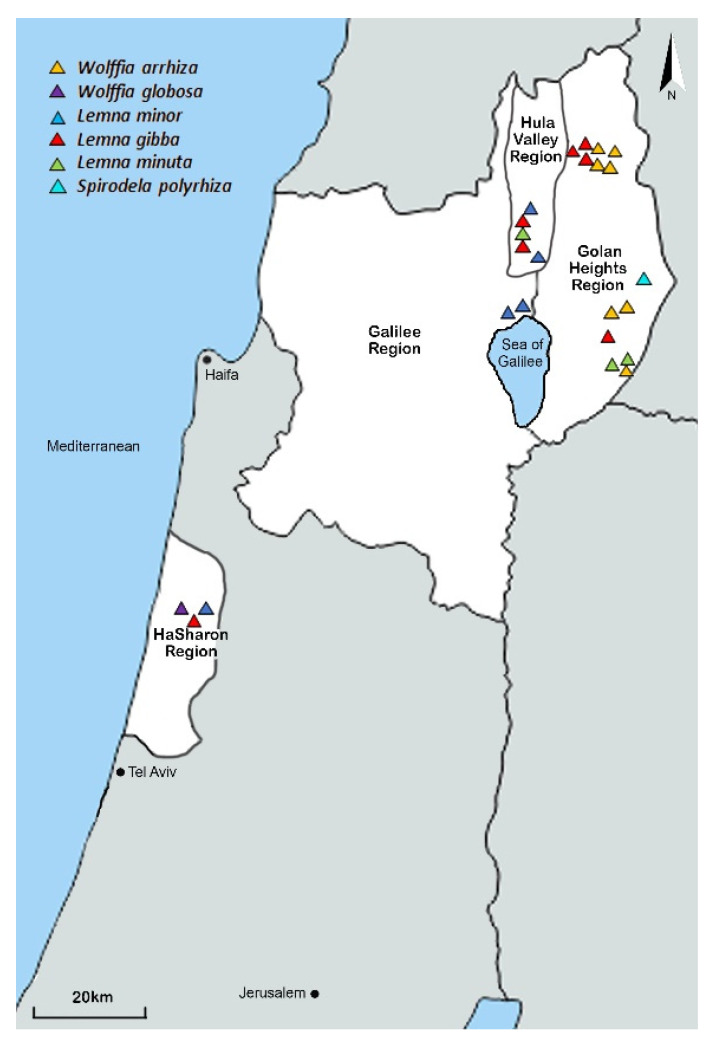
Confirmed duckweed sightings in northern and central Israel. White paint indicates the areas where the survey was conducted. Scale bar is 1:20,000.

**Figure 2 plants-11-03326-f002:**
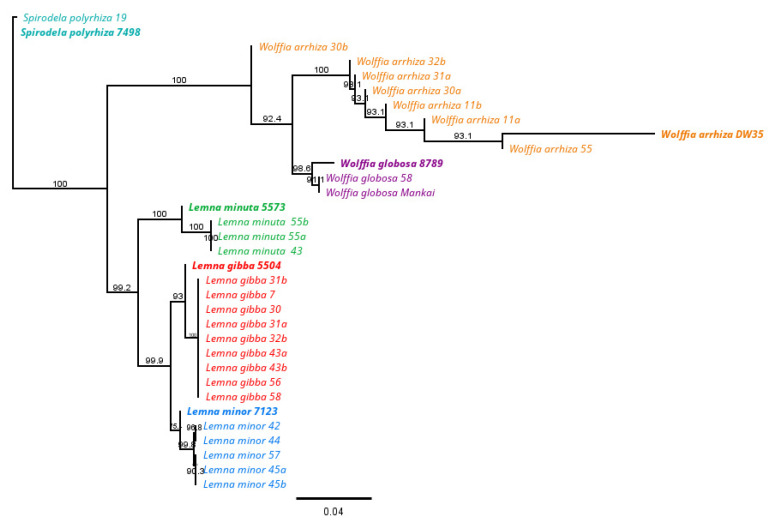
Neighbor-Joining tree using the concatenated *psbK*-*psbI* and *atpF*-*atpH* sequences and the Tamura-Nei distance. Based on 1000 reiterations. The numbers on the nodes represent the percentage of bootstrap values. Horizontal bars indicate genetic distances. Reference sequences were retrieved from the NCBI database and highlighted in bold.

**Figure 3 plants-11-03326-f003:**
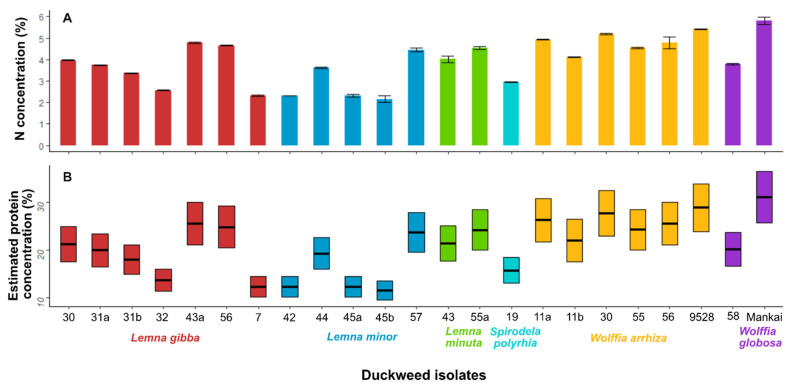
Nitrogen concentration (**A**) and estimated protein concentration (**B**) adjusted to duckweed dry biomass, which is calculated by multiplying nitrogen concentration by 4.4–6.25. The strain *W. arrhiza* 9528 was added to the analysis as a reference.

**Table 1 plants-11-03326-t001:** Identification characteristics of duckweed species found in this study [1,2]. Images were taken using an epi-fluorescent microscope.

Species	Location	No of Strains	Morphology	General Occurrence	Micrograph
*Wolffia arrhiza*	Golan Heights	9	0.5–1.5 mm long, 0.4–1.2 mm wide; ellipsoid to spherical; upper surface is convex, opaque, bright green, with its greatest width slightly below the water surface; no veins; 30–100 stomata; no roots.	Widely distributed in temperate regions; native to Europe, South Africa; invasive in Brazil, Japan, and North America.	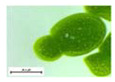
*Wolffia globosa*	HaSharon	2	0. 4–0.9 mm long, 0.3–0.6 mm wide; ellipsoid; upper surface convex, translucent pale green, with its greatest width well below the water surface; no veins; 8–25 stomata; no roots.	Tropical, subtropical, and warm temperate regions; native to eastern and southeast Asia and Africa; invasive in North America.	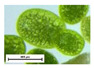
*Lemna gibba*	Golan Heights, Hula Valley, HaSharon	9	1–8 mm long, ~3.5 mm wide; lower surface of the fronds is usually gibbous; 4–5 veins extending from the nodes; >100 stomata; 1 root; difficult to identify due to high polymorphism.	Worldwide except Australia	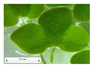
*Lemna minor*	Galilee, Hula Valley, HaSharon	5	1–10 mm long, 6–7 mm wide; upper surface shiny green, occasionally reddish; usually 3 veins, rarely 4–5; >100 stomata; 1 root.	Cooler oceanic regions; native to North America, Europe, Africa, and Western Asia.	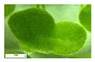
*Lemna minuta*	Golan Heights, Hula Valley	3	0.8–4 mm long, 0.5–2.5 mm wide; forming colonies of 2–4 fronds; circular with a slightly asymmetrical base; one vein, not very distinct; ~30 stomata; 1 root.	Temperate and subtropical regions, dry to moderately humid climate; native to America; invasive in Japan and Europe	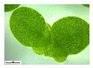
*Spirodela polyrhiza*	Golan Heights	1	Largest duckweed: 1.5–10 mm long, 1.5–8 mm wide; usually thin fronds, rarely gibbous; maximum 16 veins; >100 stomata; 7–21 roots	Worldwide	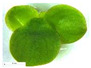

**Table 2 plants-11-03326-t002:** DNA identification of the duckweed isolates using *atpF*-*atpH* and *psbK*-*psbI* barcodes. Identification was conducted using the NCBI database.

Lab ID	Species	Region	Coordinates	Seasonality of the Water Source	*atpF*-*atpH*	Accession No	Identity (%)	*psbK*-*psbI*	Accession No	Identity (%)
					Strain Identification			Strain Identification		
32b	*L. gibba*	Golan Hights	33.137779, 35.725382	seasonal	RDSC 5504	KX212889.1	100	DW102	OM569589.1	100
58	*L. gibba*	HaSharon	32.333731, 34.876507	perennial	RDSC 5504	KX212889.1	100	DW102	OM569589.1	100
56	*L. gibba*	Golan Hights	33.09779, 35.81729	seasonal	RDSC 5504	KX212889.1	100	DW102	OM569589.1	100
7	*L. gibba*	Golan Hights	32.867917, 35.770194	seasonal	RDSC 5504	KX212889.1	100	DW102	OM569589.1	100
43b	*L. gibba*	Hulla Vally	33.060001, 35.615137	seasonal	RDSC 5504	KX212889.1	100	DW102	OM569589.1	100
43a	*L. gibba*	Hulla Vally	33.060002, 35.615138	seasonal	RDSC 5504	KX212889.1	100	DW102	OM569589.1	100
31a	*L. gibba*	Golan Heights	33.139682, 35.733806	perennial	RDSC 5504	KX212889.1	100	DW102	OM569589.1	100
31b	*L. gibba*	Golan Heights	33.139682, 35.733806	perennial	RDSC 5504	KX212889.1	100	DW102	OM569589.1	100
30	*L. gibba*	Golan Heights	33.138450, 35.734019	seasonal	RDSC 5504	KX212889.1	100	DW102	OM569589.1	100
45a	*L. minor*	Galille	32.912915, 35.569178	perennial	K46	OM569601.1	100	K46	OM569540.1	100
42	*L. minor*	Hulla Vally	33.015434, 35.629857	seasonal	K46	OM569601.1	100	K46	OM569540.1	100
44	*L. minor*	Hulla Vally	33.064187, 35.610817	seasonal	K46	OM569601.1	100	K46	OM569540.1	100
57	*L. minor*	HaSharon	32.363913, 34.958369	perennial	K46	OM569601.1	100	K46	OM569540.1	100
45b	*L. minor*	Galille	32.912915, 35.569178	perennial	K46	OM569601.1	100	K46	OM569540.1	100
55a	*L. minuta*	Golan Hights	32.801403, 35.783032	seasonal	5573	MK516255.1	100	5573	MK516236.1	100
55b	*L. minuta*	Golan Hights	32.801403, 35.783032	seasonal	5573	MK516255.0	100	5573	MK516236.2	100
43s	*L. minuta*	Hulla Vally	33.060002, 35.615138	seasonal	5573	MK516255.1	100	5573	MK516236.3	100
19	*S. polyrhiza*	Golan Hights	32.969559, 35.820036	seasonal	7498	MN419335.1	100	RDSC 2014	OM569580.1	100
58	*W. globosa*	HaSharon	32.333731, 34.876507	perennial	DW2101-4	KJ630544.1	100	5514	MG812327.1	100
11b	*W. arrhiza*	Golan Heights	32.894030, 35.775695	seasonal	DW35	OM569550.1	100	DW35	OM569611.1	99.01
30a	*W. arrhiza*	Golan Heights	33.138450, 35.734019	seasonal	DW35	OM569550.1	100	DW35	OM569611.1	99.01
31b	*W. arrhiza*	Golan Heights	33.139682, 35.733806	perennial	DW35	OM569550.1	100	DW35	OM569611.1	99.01
55	*W. arrhiza*	Golan Heights	32.801403, 35.783032	seasonal	DW35	OM569550.1	100	DW32	OM569610.1	99.29
32b	*W. arrhiza*	Golan Heights	33.137779, 35.725382	seasonal	DW35	OM569550.1	100	DW35	OM569611.1	99.31
30b	*W. arrhiza*	Golan Heights	33.138451, 35.734018	seasonal	DW35	OM569550.1	100			
11a	*W. arrhiza*	Golan Heights	32.895829, 35.776775	seasonal	DW35	OM569550.1	100	DW35	OM569611.1	99.31

**Table 3 plants-11-03326-t003:** Fatty acid composition and content of duckweed plants collected in the current study. The strain *W. arrhiza* 9528 was added to the analysis as a reference. An isomer of 16:1 and 18:1*n*7 are not shown. The full table is shown in the Appendix A.

	Fatty Acid (% of Total Fatty Acids)													ω3/ω6 18:3/18:2	TFA (%DW)
Haplotype	16:0	16:1	16:2	16:3	18:0	18:1n9	18:2n6	18:3n6 GLA	18:3n3 ALA	18:4n3 SDA	20:0	22:0	24:0		
*L. gibba* 58	20.56 ± 0.43	3.91 ± 0.42	2.2 ± 0.05	0.52 ± 0.13	0.92 ± 0.04	1.13 ± 0.13	11.1 ± 0.33	0.61 ± 0.05	52.88 ± 0.65	2.62 ± 0.08	0.37 ± 0.01	0.44 ± 0.04	1.13 ± 0.03	4.77	5.25
*L. gibba* 30	19.72 ± 0.46	4.88 ± 0.03	3.27 ± 0.05	0.75 ± 0.04	1.01 ± 0.16	1.07 ± 0.09	10.46 ± 0.17	0.89 ± 0.05	50.32 ± 0.36	3.81 ± 0	0.45 ± 0.01	0.39 ± 0.02	1.15 ± 0.04	4.81	4.39
*L. gibba* 31a	21.32 ± 0.05	4.28 ± 0.04	2.06 ± 0.01	0.17 ± 0.02	1.25 ± 0.01	0.78 ± 0	11.27 ± 0.18	0.72 ± 0	50.26 ± 0	4.37 ± 0.03	0.53 ± 0	0.57 ± 0.01	1.05 ± 0.01	4.46	3.63
*L. gibba* 31b	21.61 ± 0.17	5.46 ± 0.31	2.18 ± 0.02	0.23 ± 0.07	1.07 ± 0.04	1.11 ± 0.01	13.93 ± 0.05	0.39 ± 0	48.98 ± 0.01	1.96 ± 0.03	0.39 ± 0	0.41 ± 0.01	0.84 ± 0.02	3.52	3.88
*L. gibba* 32b	20.01 ± 0.42	4.65 ± 1.09	2.11 ± 0.03	0.35 ± 0.21	1.36 ± 0.05	0.94 ± 0.01	15.37 ± 0.09	0.8 ± 0.01	48.09 ± 0.21	3.21 ± 0.01	0.55 ± 0.02	0.51 ± 0.05	0.93 ± 0.02	3.13	4.10
*L. gibba* 43a	20.96 ± 0.23	3.69 ± 0.42	2.07 ± 0.06	0.65 ± 0.08	1.01 ± 0.08	1.47 ± 0.24	13.83 ± 0.02	0.61 ± 0.07	49.87 ± 0.91	2.06 ± 0.14	0.48 ± 0.02	0.51 ± 0.01	1.1 ± 0.19	3.60	4.36
*L. gibba* 43b	21.4 ± 0.31	4.56 ± 0.77	1.96 ± 0.03	0.43 ± 0.13	0.97 ± 0.01	1.3 ± 0.02	13.69 ± 0.07	0.47 ± 0	49.92 ± 0.24	2.02 ± 0	0.39 ± 0.01	0.41 ± 0.02	0.76 ± 0.03	3.65	4.12
*L. gibba* 56	21.75 ± 2.27	5.39 ± 1.1	2.62 ± 0.02	0.23 ± 0.11	1.08 ± 0.31	1.08 ± 0.19	13.82 ± 1.37	0.52 ± 0.11	47.77 ± 3.37	2.35 ± 0.34	0.42 ± 0.14	0.45 ± 0.15	0.71 ± 0.2	3.49	4.09
*L. gibba* 7	21.49 ± 0.04	4.23 ± 0.22	2.24 ± 0.06	0.4 ± 0.03	1.24 ± 0.03	0.94 ± 0.01	15.3 ± 0.06	0.44 ± 0.05	47.78 ± 0.51	2.1 ± 0.03	0.53 ± 0.01	0.5 ± 0.01	1.4 ± 0.13	3.12	2.83
*L. minor* 42	22.73 ± 0.12	5.67 ± 0.37	1.62 ± 0.09	0.35 ± 0.06	1.8 ± 0.12	1.13 ± 0.02	16.01 ± 0.08	0.46 ± 0.01	45.8 ± 0.03	1.51 ± 0.14	0.55 ± 0	0.31 ± 0	1.01 ± 0.01	2.86	3.35
*L. minor* 44	19.85 ± 0.06	5.5 ± 0.01	2.01 ± 0.01	0.2 ± 0.01	1.28 ± 0.02	1.45 ± 0	18.4 ± 0.1	0.5 ± 0	46.75 ± 0.13	1.75 ± 0	0.31 ± 0.01	0.29 ± 0.01	0.82 ± 0.04	2.54	6.34
*L. minor* 45a	19.97 ± 0.33	2.89 ± 0.05	2.03 ± 0.02	0.75 ± 0.02	1.4 ± 0.04	1.69 ± 0.01	18.88 ± 0.05	0.6 ± 0.01	47.37 ± 0.39	1.88 ± 0.08	0.47 ± 0.01	0.36 ± 0.15	0.72 ± 0.78	2.51	3.77
*L.minor* 45b	22.49 ± 0.9	4.1 ± 0.79	2.54 ± 0.43	0.68 ± 0.37	1.4 ± 0.16	1.57 ± 0.03	17.64 ± 0.15	0.71 ± 0.05	42.74 ± 0.63	2.68 ± 0.21	0.45 ± 0.01	0.34 ± 0.14	1.16 ± 0.15	2.42	3.52
*L. minor* 57	18.28 ± 0.16	5.8 ± 0.18	2.77 ± 0.02	0.22 ± 0.04	0.77 ± 0.02	1.33 ± 0.01	16.1 ± 0.11	1.27 ± 0.01	48.36 ± 0.19	3.04 ± 0.01	0.17 ± 0.01	0.24 ± 0	0.71 ± 0.03	3.00	4.19
*L. minuta* 43	18.48 ± 0.45	4.64 ± 0.03	2.43 ± 0.05	0.52 ± 0.05	1.24 ± 0.39	1.27 ± 0.02	14.07 ± 0.21	0.23 ± 0.21	54.43 ± 1.13	0.09 ± 0.02	0.25 ± 0.08	0.29 ± 0.05	1.13 ± 0.1	3.87	4.78
*L. minuta* 55a	19.92 ± 0.2	4.97 ± 0.52	1.62 ± 0	0.31 ± 0.13	0.93 ± 0.03	1.08 ± 0.02	15.87 ± 0.14	0 ± 0	53.35 ± 0.23	0.12 ± 0.11	0.15 ± 0.02	0.19 ± 0.03	0.79 ± 0.06	3.36	3.95
*L. minuta* 55b	20.07 ± 0.12	3.44 ± 0.07	1.6 ± 0	0.66 ± 0.04	1.03 ± 0.02	1.35 ± 0.07	15.91 ± 0.07	0 ± 0	53.51 ± 0.09	0.13 ± 0.18	0.25 ± 0	0.15 ± 0	1.17 ± 0.02	3.36	4.19
*S. polyrhiza* 19	23.15 ± 0.24	5.43 ± 0.37	6.11 ± 0.06	0.35 ± 0.04	2.35 ± 0.07	1.17 ± 0.04	5.23 ± 0	0.04 ± 0.05	50.9 ± 0.48	0 ± 0	0.58 ± 0.04	0.62 ± 0.01	2.61 ± 0.05	9.73	4.27
*W. arrhiza* 11a	23.22 ± 1.1	5.64 ± 0.58	3.64 ± 0.23	0.55 ± 0.03	1.62 ± 0	2.17 ± 0.08	25.24 ± 0.32	0.28 ± 0.01	34.87 ± 0.64	0.1 ± 0.01	0.77 ± 0.01	0.64 ± 0.04	0.4 ± 0.03	1.38	4.38
*W. arrhiza* 11b	21.63 ± 0.26	5.36 ± 0.53	3.45 ± 0.12	0.59 ± 0.11	1.48 ± 0.07	1.87 ± 0.03	25.7 ± 0.02	0.15 ± 0.01	37.32 ± 0.56	0.1 ± 0.01	0.73 ± 0.01	0.53 ± 0.01	0.37 ± 0.08	1.45	4.55
*W. arrhiza* 30a	20.4 ± 0.01	5.32 ± 0.2	3.46 ± 0.02	0.48 ± 0	1.17 ± 0.02	1.73 ± 0.01	22.75 ± 0.02	0.14 ± 0	42.29 ± 0.14	0.08 ± 0	0.58 ± 0	0.47 ± 0.01	0.31 ± 0.02	1.86	4.70
*W. arrhiza* 31b	24.28 ± 0.14	3.11 ± 0.22	3.19 ± 0.14	1.03 ± 0.14	1.35 ± 0.04	1.64 ± 0.04	23.78 ± 0.03	0.11 ± 0.1	38.74 ± 0.33	0.14 ± 0.02	0.69 ± 0.02	0.4 ± 0	0.52 ± 0.03	1.63	3.18
*W. arrhiza* 55	21.85 ± 0.43	3.65 ± 0.25	3.33 ± 0.04	0.75 ± 0.04	1.37 ± 0.11	1.73 ± 0.04	25.7 ± 0.26	0.03 ± 0.01	38.97 ± 0.44	0.1 ± 0.01	0.6 ± 0.02	0.51 ± 0.04	0.34 ± 0.08	1.52	5.13
*W. arrhiza* 9528	19.05 ± 0.13	3.72 ± 0.46	2.48 ± 0.03	0.24 ± 0.02	1.81 ± 0.01	1.79 ± 0.01	24.29 ± 0.12	0.14 ± 0.01	44.41 ± 0.1	0.07 ± 0	0.86 ± 0.02	0.66 ± 0.05	0.29 ± 0.02	1.83	4.44
*W. globosa* 58	22.71 ± 0.05	4.04 ± 0.58	1.5 ± 0.05	0.35 ± 0.08	2.25 ± 0	2.41 ± 0.01	25.02 ± 0.01	0.07 ± 0.39	39.51 ± 0.04	0.02 ± 0.02	0.53 ± 0.01	0.39 ± 0.02	0.19 ± 0.01	1.58	4.38
*W. globosa* Mankai	21.73 ± 0.16	4.39 ± 0.64	2.34 ± 0.02	0.74 ± 0.21	2.14 ± 0.02	1.9 ± 0.01	19.88 ± 0.27	0.14 ± 0	44.64 ± 0.21	0.07 ± 0	0.41 ± 0.02	0.19 ± 0.01	0.33 ± 0.02	2.25	5.36

## Data Availability

Not Applicable.

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
