# Peer review of "Diversity and Differentiation of Duckweed Species from Israel"

_plants, 2022, doi:10.3390/plants11233326_

Round 1

Reviewer 1 Report

The article addresses a timely issue because the location under investigation is unique for the processes of duckweed distribution by birds.
As a whole, the article is written at a high scientific level; however, there are several remarks that will help improve the understanding of the presented material.

Abstract. Please add the relevance and purpose of the work (the first and second sentences).

There are no conclusions in the article. Please add them.

Move Section Materials and Methods before the Results section so that the reader can understand the procedure for conducting the experiment before reading the main results.

Author Response

Reply to Reviewer 1.

Point 1: Abstract. Please add the relevance and purpose of the work (the first and second sentences).

Response 1: Added

Point  2: There are no conclusions in the article. Please add them.

Response 2: A conclusion section was added (line 408)

Point 3: Move Section Materials and Methods before the Results section so that the reader can understand the procedure for conducting the experiment before reading the main results

Response 3: The Materials and Methods section was moved (line 123)

Reviewer 2 Report

The manuscript “Diversity and differentiation of duckweed species from Israel” records duckweeds diversity in a special geographic region connecting Europe, Africa and Asia. For species identification the authors used molecular tool of chloroplast DNA barcoding, complemented with monitoring amounts of nitrogen/protein and content/composition of fatty acids.   The study is well written and illustrated, providing a solid addition to our knowledge about duckweed biodiversity in a poorly explored key inter-continental location.

The minor isuues to address for manuscript’s improvement:

Line 43-45: “Three of the genera contain one or few tiny roots emerging from the fronds (Spirodela, Landoltia and Lemna), while the two remaining genera are rootless and smaller”; better to write “Representatives of three genera contain….”

Line 87: “The study the duckweed diversity…” change to “The study of duckweed diversity…”

Line 96: “Never the less…”  to “Nevertheless…”

Lines 100-103: The sentence should be corrected.

Lines 108-116: Make species names italic.

Line 149: “There were four major fatty acids…”;  three, not four, FAs are specified.

Line 175-176: “The other W. globosa, an isolate, showed lower concentrations”. The meaning is not clear.

The study is based on the species identification using DNA barcoding, however it is totally missing in discussion. I would recommend adding some information on the subject.  

The website link https://biogis.huji.ac.il/heb/home.html is mentioned 4 times in the first three paragraphs of discussion and 3 times in lines 275-285; a bit too much, I guess.  

Author Response

Response to Reviewer 2

Line 43-45: “Three of the genera contain one or few tiny roots emerging from the fronds (Spirodela, Landoltia and Lemna), while the two remaining genera are rootless and smaller”; better to write “Representatives of three genera contain….”

Response 1: Done.

Line 87: “The study the duckweed diversity…” change to “The study of duckweed diversity…”

Response 1: Done.

Line 96: “Never the less…”  to “Nevertheless…”

Response 1: Done.

Lines 100-103: The sentence should be corrected.

Response 1: Done.

Lines 108-116: Make species names italic.

Response 1: Done.

Line 149: “There were four major fatty acids…”;  three, not four, FAs are specified.

Response 1: fixed

Line 175-176: “The other W. globosa, an isolate, showed lower concentrations”. The meaning is not clear.

Response 1: The sentence was edited for better understanding.

The study is based on the species identification using DNA barcoding, however it is totally missing in discussion. I would recommend adding some information on the subject.  

Response 1: A paragraph dealing with identification by molecular barcoding has been added to the discussion (lines 311-320).

The website link https://biogis.huji.ac.il/heb/home.html is mentioned 4 times in the first three paragraphs of discussion and 3 times in lines 275-285; a bit too much, I guess.  

Response 1: It has been reduced to two mentions of the website.